# Physiological Predictors of Performance on the CrossFit “Murph” Challenge

**DOI:** 10.3390/sports8070092

**Published:** 2020-06-28

**Authors:** Ja’Deon D. Carreker, Gregory J. Grosicki

**Affiliations:** Biodynamics and Human Performance Center, Georgia Southern University (Armstrong Campus), Savannah, GA 31419, USA; jadeoncarreker@gmail.com

**Keywords:** CrossFit, Murph, body fat, strength, fitness, Wingate, VO_2_max

## Abstract

We examined physiological predictors of performance on the CrossFit Murph challenge (1-mile run, 100 pullups, 200 pushups, 300 air squats, 1-mile run). Male CrossFit athletes (*n* = 11, 27 ± 3 years) performed a battery of physical assessments including: (1) body composition, (2) upper and lower body strength, (3) upper body endurance, (4) anaerobic power, and (5) maximal oxygen consumption. No less than 72 h later, participants completed the Murph challenge, heart rate was monitored throughout, and blood lactate was obtained pre-post. Correlations between physiological parameters and total Murph time, and Murph subcomponents, were assessed using Pearson’s correlations. Murph completion time was 43.43 ± 4.63 min, and maximum and average heart rate values were 185.63 ± 7.64 bpm and 168.81 ± 6.41 bpm, respectively, and post-Murph blood lactate was 10.01 ± 3.04 mmol/L. Body fat percentage was the only physiological parameter significantly related to total Murph time (r = 0.718; *p* = 0.013). Total lift time (25.49 ± 3.65 min) was more strongly related (r = 0.88) to Murph time than total run time (17.60 ± 1.97 min; r = 0.65). Greater relative anaerobic power (r = −0.634) and less anaerobic fatigue (r = 0.649) were related to total run time (*p* < 0.05). Individuals wanting to enhance overall Murph performance are advised to focus on minimizing body fat percentage and improving lift performance. Meanwhile, performance on the run subcomponent may be optimized through improvements in anaerobic power.

## 1. Introduction

CrossFit is defined as “a high-intensity fitness program incorporating elements from several sports and types of exercise” [1]. The sport of CrossFit is a rapidly-expanding genre of fitness, reaching approximately 15,000 affiliates worldwide [2]. The foundation of CrossFit prides itself upon constantly varied, high intensity functional movements to improve fitness and health. Each CrossFit workout is designed to task the body with significant stimuli to produce meaningful mechanical and metabolic adaptations. The Murph challenge is highly-celebrated within the CrossFit community, receiving its name from Lt. Michael P. Murphy, who was killed in Afghanistan in 2005 and received a Medal of Honor posthumously. The Murph starts with a 1-mile run, followed by 100 pullups, 200 pushups, and 300 squats, and finishes with another 1-mile run, and is traditionally completed wearing a 20-Ib vest. The pullups, pushups, and squats can be done in any order and the objective of the challenge is to complete the sequence as fast as possible. The diverse nature of these tasks stresses a variety of physiological competencies (e.g., metabolic, mechanical, etc.).

Metabolically, energy may be produced via anaerobic or aerobic means, with exercise duration and intensity being the greatest determinants of energy systems reliance. Specifically, anaerobic energy systems are favored during high-intensity exercise lasting < 120 s, after which aerobic metabolism predominates. Time to completion of the Murph usually ranges from 22 to 60 min [3], and thus Murph completion is largely reliant on aerobic metabolism. However, the intermittent nature of the lift session that involves transitioning between pushups, pullups, and air squats may be influenced by anaerobic performance and the ability to recover from high intensity exercise [4]. As such, inter-individual differences in the ability to perform anaerobic work and manage hydrogen ion accumulation [5], as well as aerobic power, may help to explain performance on the Murph challenge.

Mechanical elements such as muscular strength and endurance may also influence Murph performance. Muscular strength is defined as “the maximum force-generating capacity of a muscle or group of muscles” [6], while muscular endurance is defined the ability of a muscle or muscle group to perform repeated contractions against a load for an extended period of time [6]. It is possible that maximal strength has a positive correlation with consecutive repetitions when using the same muscle groups at a sub-maximal level [7]. The importance of muscular endurance pertaining to the Murph is justified through specificity. The lift session requires the muscles of the chest, back, and legs to fire repeatedly until the repetition goal is met for each movement.

Unlike many CrossFit workouts, the Murph does not require any external weight aside from an optional 20-pound vest. Instead, the Murph employs bodyweight exercises and thus inter-individual differences in body composition may relate to performance time. Body composition can be examined in many ways. The Body Mass Index (BMI; kg/m^2^) is the most common but faces the limitation of not considering proportions of muscle and fat. Body fat percentage (%BF) can be estimated through a variety of methods, and each technique has its unique pros and cons. Estimates of %BF can be attained using the skinfold method, hydrostatic weighing, or dual-energy X-ray absorptiometry (DEXA), which is the gold standard due to its ability to distinguish among muscle, bone, and fat. 

Previous research has queried the contribution of the different fitness parameters described above (i.e., aerobic and anaerobic power, muscular strength and endurance, and body composition) to CrossFit workout performance [8,9,10,11]. However, to date there has been no research comparing the aforementioned fitness parameters with performance on the Murph challenge, one of the most popular CrossFit workouts. The objective of this study was to distinguish which physiological parameters most strongly influence Murph performance. Due to the scientific novelty of the Murph challenge, we also aimed to characterize cardiovascular (heart rate) and metabolic (blood lactate) responses to performing the Murph. We hypothesized that aerobic power (VO_2_max) would be the greatest predictor of Murph performance, and that the substantial physiological demand of completing the Murph would be exhibited by near maximal heart rate values and substantial blood lactate accumulation.

## 2. Materials and Methods

### 2.1. Participants

Eleven healthy, active young (18–40 years) men with at least 6 months of CrossFit experience (at least twice a week) volunteered to participate in the study, all of whom self-reported at least one completion of the Murph. All interested individuals were given a full description of study procedures and provided written consent to participate. All participants were free from acute or chronic illness (e.g., cardiac, pulmonary, liver, or kidney abnormalities, cancer, hypertension, diabetes, or other known metabolic disorders), free from orthopedic limitations, not taking any heart-rate altering medications, and they did not smoke or participate in other forms of tobacco use. 

The protocol was approved by the Institutional Review Board at Georgia Southern University. The study consisted of two testing visits, before which participants were asked to refrain from vigorous physical activity for 48 h and report to the testing facility in a hydrated state and having eaten their last meal ~2–3 h prior. Upon the first visit, participants were then taken through a comprehensive physiological screening battery including the following measures which are described in greater detail below: body composition, upper and lower body muscular strength, muscular endurance, and finally anaerobic and aerobic power. 

### 2.2. Body Composition

Body composition was measured via dual-energy X-ray absorptiometry (DEXA) (Lunar iDXA, GE Healthcare, Madison, WI, USA). The DEXA machine was outfitted with enCORE version 16 and the machine was calibrated prior to each scan as per manufacturer instructions (laboratory coefficient of variation < 0.07%). During the test, subjects laid supine and remained motionless on the examination table for 5–10 min. The information acquired from the DEXA included body fat percentage, total mass, lean mass, fat mass, bone mineral density, and bone mineral content.

### 2.3. Muscular Strength

Upper and lower body muscular strength, as one-repetition maximum (1RM), were evaluated using the bench press and back squat exercises, respectively. Before beginning the assessment, participants were familiarized with each exercise to ensure proper form and technique. Using a light weight (estimated 50% 1RM), participants completed 10 repetitions. After 3–5 min of rest, the weight was increased to an estimated 75% of maximum and participants were asked to complete a single repetition. The weight was increased by 5–10% and the participant completed another single repetition. This process was repeated until the individual was no longer able to complete a repetition. The maximal amount of weight with which the individual was able to successfully complete a repetition throughout the entire range of motion was quantified as the 1RM. Ten minutes of rest was provided between upper- (bench press) and lower- (squat) body assessments. The same protocol was used for the back-squat exercise. As previously described, upper- and lower body muscular strength were combined to make strength total [9] and relative strength was computed as strength total/body mass (kg).

### 2.4. Upper-Body Endurance

Upper-body endurance was quantified as the greatest number of repetitions that the participant was able to complete, through the full range of motion on the bench press exercise, using a load corresponding to 50% of their 1RM [12]. Ten minutes after evaluation of muscular strength, the appropriate load was calculated, and participants were asked to complete as many repetitions as possible while maintaining good technique throughout. If the technique was altered, or a repetition was failed, the test was terminated by the spotter.

### 2.5. Wingate Anaerobic Test

Anaerobic power was quantified using the Wingate anaerobic testing (WnAT) protocol. After explaining the testing protocol, participants were taken to the testing ergometer (Monark, 894e, Vansbro, Sweden) and handlebar position and seat height (~155-degree angle behind knee) were fitted to each participant to maximize safety and comfort. Individuals were given a 4-min warm-up where they were asked to pedal at a light load (~30 W). The load was then totally removed, and the participants were asked to pedal “all-out” achieving the highest possible pedaling frequency of which they were able. Once the pedaling frequency was reached, the participant pressed a button which automatically reapplied the load (7.5% body mass). Participants then provided their best effort to sustain the highest possible pedaling frequency for 30 s. Once 30 s was reached, the load was removed, and participants cooled down for 3 min at a light load (30 W). Variables gathered from this assessment include (a) absolute and relative peak power, (b) absolute and relative peak power, (c) anaerobic fatigue ((peak power − minimum power) / peak power (×100)), and (d) total work. Relative power was calculated as work (W) divided by body mass in kg. Participants were given a 10-min rest period before moving on to cardiorespiratory fitness evaluation.

### 2.6. Cardiorespiratory Fitness (VO_2_Max)

Aerobic power (VO_2_max) was assessed using a 4Front (Woodway, Waukesha, WI, USA) motorized treadmill using a standardized graded testing protocol. The test started with a two-minute walking (3 mph) warm-up, after which speed was increased to 5 mph. Every two minutes thereafter, the speed was increased by 2 mph until a rating of perceived exertion of 13 (Borg 6–20 scale) or greater was reported. For the next testing stage, grade was increased to 4% and every two minutes thereafter grade was increased by 2% per stage until volitional exhaustion was achieved. This protocol is consistently used in our laboratory in this population (i.e., young healthy men) due to its known ability to elicit fatigue within 8–12 min, as per the recommendation of the American College of Sports Medicine [13]. Heart rate (HR, via wireless telemetry) (Polar, H10, Bethpage, NY, USA) and ratings of perceived exertion (RPE) were recorded in the last 15–20 s of each testing stage. Attainment of VO_2_max was confirmed by satisfying two of the three following criteria: Respiratory Exchange Ratio (RER) > 1.1, RPE > 17, and/or achievement of 90% age-predicted maximum heart rate. Verbal encouragement was provided in a standardized manner throughout the test. Oxygen uptake was measured and averaged in 15-s intervals and VO_2_max was classified as the highest average of two consecutive readings. Maximal heart rate was defined as heart rate peak. Immediately after the test, participants were given a two-minute cooldown period where they were asked to walk at 3 mph.

### 2.7. Visit 2

The second testing visit was scheduled at least 72 h after the first physiological assessment battery. On the day of the second testing visit, participants were asked to perform the Murph challenge, as quickly as possible. The treadmill and platform for the lifting section of the Murph was separated by 3 m. Briefly, subjects completed a 1-mile treadmill run (at a self-selected pace that participants were able to modify throughout) followed by 10 sets of 10 pullups, 20 pushups, and 30 air squats, immediately followed by another 1-mile treadmill run. If there came a time when participants could not complete the 10-20-30 set/repetition scheme, the repetitions were partitioned (i.e., 5 pullups, 10 pushups, 15 air squats) until the goal number of repetitions were met. This repetition scheme was chosen as per the recommendation of multiple CrossFit facilities and coaches. When performing the pullups, participants were able to use the strict or kipping technique. For each pullup repetition, the study investigator ensured that the participant started with adequate elbow extension and finished with their chin above the bar. Heart rate was monitored throughout the testing session via wireless telemetry in order to characterize the heart rate response to the Murph challenge. Immediately before and then again three minutes after completion of the challenge, blood lactate was evaluated via fingerstick (Lactate Scout+, EFK Diagnostics, Cardiff, UK) in order to assess the metabolic implications of performing the Murph.

### 2.8. Statistical Analysis

Statistical analyses were performed using Statistical Package for the Social Sciences (IBM SPSS, version 25, Armonk, NY, USA). Participant characteristics were calculated as means ± SD. Normality of data were confirmed using Shapiro Wilk’s test and boxplots. Simple Pearson’s r correlations were used to assess associations between Murph completion time and its subcomponents (i.e., run time and lift time), and the physiological measures. A simple linear regression model was then created using the significant correlative data. Statistical significance was set at an alpha level of 0.05.

## 3. Results

### 3.1. Participant Characteristics 

Figure 1 is a CONSORT diagram showing recruitment efforts. Participant characteristics are presented in Table 1. Participants’ age ranged from 21–31 years. Participants reported ~4 years of CrossFit experience, with a substantial range from 13 to 120 months. Descriptive statistics for the physiological parameters can be found in Table 2. DEXA-derived body fat percentage fell into the 60th percentile compared to age- and gender-matched norms [14]. According to the American College of Sports Medicine (ACSM), the average VO_2_max of the participants was within the 55th percentile for their age group when compared to general population [13]. Upper body strength assessed using bench press relative to body mass was 1.38, which is within the 85th percentile for their sex-specific age range [13].

### 3.2. Physiological Responses and Predictors of Murph

Descriptors of performance measures, physiological responses, and Murph subcomponents along with their respective correlations to Murph completion time are shown in Table 2. On average, participants’ blood lactate increased to approximately 10 mmol/L. Average heart rate was 168 bpm, which was ~85% of maximal values witnessed during VO_2_max testing. The participants’ time to completion of the Murph was 43.43 ± 4.63 min and ranged from 36.56 to 54.21 min. All but three participants were able to complete the lift portion of the workout using the 10-20-30 repetition scheme. No apparent trends (advantages or disadvantages) were observed in the individuals who dropped to the 5-10-15 repetition scheme and thus all data were analyzed together.

Correlational results indicated that body fat percentage was the only variable significantly (*p* < 0.05) related to Murph completion time (Figure 2). When entered into the regression model, a significant regression equation was found (F(1,8) = 8.444, *p* = 0.020), with an adjusted R2 of 0.453. Participants’ predicted Murph time is equal to 28.816 + 0.822 (body fat percentage) min. Participants’ Murph time increased 49 s for each percent of body fat. Total lift time was more strongly related (r = 0.88) to Murph completion time than was total run time (r = 0.652), but both were significant predictors (*p* < 0.05; Figure 3).

### 3.3. Physiological Predictors of Murph Subcomponents

Correlations between physiological variables and Murph subcomponents (i.e., run and lift times) are shown in Table 3. Body fat percentage was the only parameter significantly related to both run and lift times (*p* < 0.05). Lower anaerobic fatigue was correlated (*p* < 0.05) with better performance on run one and total run time. Relative average power was significantly correlated (*p* < 0.05) with total run time. Intriguingly, neither absolute nor relative VO_2_max were significantly related (*p* > 0.05) to any of the run times.

## 4. Discussion

### 4.1. Summary

The primary purpose of this study was to determine the physiological predictors of success for the CrossFit Murph challenge. Participants within this study were physiologically robust, consistently demonstrating aerobic and anaerobic competencies that were comparably superior to the average population [13]. Interestingly, the only physiological parameter that significantly correlated with Murph completion time was total body fat percentage, and not cardiorespiratory fitness (VO_2_max), as hypothesized. Examination of Murph subcomponents (i.e., total run and total lift time) demonstrated that lift time was more strongly associated with total Murph completion time than run performance. A novel aspect of the present investigation was that we also examined physiological responses to the Murph challenge. Heart rate and blood lactate responses observed during and after the Murph challenge highlight the high degree of cardiovascular and metabolic stress placed upon the body when completing the Murph. Average heart rate indicates that throughout the workout, participants were exercising at approximately 85% of their heart rate max for an extended period of time, causing rapid accumulation of blood lactate due to a substantial reliance on anaerobic energy-producing means. Collectively, these findings provide valuable information to CrossFit athletes by highlighting the importance of body composition and the lift portion of the Murph as key variables in overall Murph success.

### 4.2. Predictors of Murph Performance

Body fat percentage explained approximately half of the variance in Murph completion time. Throughout previous literature, body fat percentage has been shown to be significantly correlated with physical performance such as muscular strength, power, endurance, and anaerobic capacity [15,16,17,18,19]. Fat is noncontractile and does not contribute to force production, increasing the force and energy requirements of the muscles, particularly for bodyweight movements [19]. With the requirements of the Murph being entirely comprised of bodyweight exercises (pullups, pushups, air squats), non-contractile tissue mass would be anticipated to detrimentally affect performance time. Similarly, body fat percentage is a well-known predictor of running performance, as was demonstrated by a 2012 study by Barandun et al. (2012) in which body fat explained 44% of marathon race time [20]. Knechtle et al. (2014) and Tanda et al. (2013) also concluded that body fat percentage could be used to predict half marathon and marathon race times, respectively [21,22]. In fact, the importance of non-functional body mass as an impediment to Murph performance is highlighted by the fact that the traditional Murph challenge is performed with a 20-lb vest. Whether the relationship between body fat percentage and overall Murph performance would still be realized if the participants had worn a weight vest is less certain and worthy of future inquiry.

### 4.3. Sub-Components Analysis

Total run time and total lift time explained 42% and 77% of total Murph time, respectively. Interestingly, neither absolute nor relative strength, nor muscular endurance, was related to lift time. Though it may be anticipated that muscular endurance would predict lift performance, it is important to keep in mind that our measure of muscular endurance was made relative to an individual’s maximal strength. Meanwhile, during the Murph challenge muscular endurance is assessed relative to an individual’s body mass. Thus, an individual with relatively poor muscle strength may have been classified as having fairly robust muscular endurance using our measurements, yet insufficient muscular strength to lift their own body mass during the Murph challenge. Literature regarding maximal muscular strength as related to relative muscular endurance has been mixed. Dean et al. (1987) concluded that bench press strength accounted for ~50% of the variance in push-up performance, whereas Invergo et al. (1991) and Mayher et al. (1991) concluded that there was no significant relationship between push-up performance and 1RM bench press [23,24,25]. Our finding of a lack of relationship between absolute and relative strength using bench press and lift performance during the Murph challenge highlights the importance of specificity, even if theoretically training the same physiological system (i.e., muscular endurance).

Aerobic capacity did not serve the paramount role in total Murph completion time that we hypothesized. Furthermore, neither relative nor absolute VO_2_max were related to completion of first, second, or total run time. Running performance is shown to be influenced by VO_2_max, running economy, and lactate threshold [26]. Interestingly, exploratory analysis revealed that anaerobic measures including relative 30-s power and anaerobic fatigue were significantly correlated with total run time. Anaerobic fatigue, expressed as a percent decline, highlights the decline in anaerobic power throughout the 30-s Wingate test; therefore, a low anaerobic fatigue value would be favored, meaning that there was little decrement to anaerobic power. Previous research has shown that different anaerobic parameters can be used to improve running performance by improving running economy and lactate tolerance, even in distances that are relying on the aerobic system for energy production. Vorup et al. (2015) observed that, following an eight-week program of strength and speed endurance training, endurance-trained runners experienced a significant increase in time to exhaustion, maximal aerobic speed, and peak blood lactate, though there was no change in VO_2_max or pulmonary oxygen uptake at submaximal running speeds [27]. Nummela et al. (2006) observed a significant relationship between the velocity of a maximal anaerobic running test and the velocity of a 5-km time trial, concluding that distance running performance and running economy are related to neuromuscular capacity to produce force [28]. Baumann et al. (2012) reported similar results as Nummela, stating that anaerobic energy production explains a significant amount of variation seen in 5-km-race performance [29]. The ability to tap into anaerobic energy-producing means may have provided some participants with an advantage during the run portions of the Murph, highlighting the value of high-intensity training programs focused on improving lactate tolerance to improve run performance in the Murph challenge.

### 4.4. Limitations

A limitation of the study is that the Murph was performed in a controlled setting with only one subject per trial. With the Murph traditionally being performed in a CrossFit facility within a group, this decreases external validity of the study. This was a significant consideration during our study design process, but after much deliberation we chose to preference internal validity through superior controls, as previous research groups have done [8,9]. Internal validity of our findings was also preserved through the use of the specific 10-20-30 repetition scheme when performing the Murph, although relationships between physiological characteristics and Murph success may differ if a different partitioning protocol is employed. Previous Murph experience might also have affected completion times, though no relationship was observed between CrossFit experience and Murph performance (data not shown). Another possible limitation is that subjects were given a timeframe of 3 to 14 days between the two testing visits. The difference in recovery time may influence the performance of the Murph for the second visit; however, given our conservative 72 h window we felt subjects would likely be fully recovered, as was confirmed by participants in oral communication. Additionally, an assessment of lower extremity muscular endurance may have provided greater insight into predictors of Murph success. After much debate we decided not to measure lower extremity muscular endurance so as to avoid undue fatigue during the latter portion of the physiological testing battery. Indeed, even though the testing battery was administered as per the recommendations of the National Strength and Conditioning Association, performance on the Wingate and VO_2_max tests may have been impacted already. Our use of the treadmill for the running portion of the Murph may also be viewed as a limitation of the present study. Inter-individual differences in experience with running on a treadmill as well as our use of a 0% rather than 1% grade, which may more accurately reflect the energy cost of outdoor running [30], may have impacted our results. Finally, our relatively modest sample size is another undeniable limitation, although as highlighted in the CONSORT diagram (Figure 1) recruiting and enrolling participants for such an involved study was quite difficult and our sample size was not dissimilar to previous research in the field [9]. 

## 5. Conclusions

In summary, our findings demonstrate that the performance of the Murph challenge can be predicted using DEXA-derived body fat percentage. Neither metabolic (aerobic/anaerobic power) nor mechanical (muscle strength/endurance) variables exhibited a significant relationship with overall Murph time to completion. Additionally, we demonstrated that completion of the lift portion for the Murph task was of greater relevance for overall completion time than run time. Although they were not related to total Murph time, low anaerobic fatigue and high relative anaerobic power increases the chances of completing the run portions of the Murph at a faster pace. Interpreted together, these findings emphasize the value of minimizing body fat percentage and optimizing lift performance for overall Murph success.

## Figures and Tables

**Figure 1 sports-08-00092-f001:**
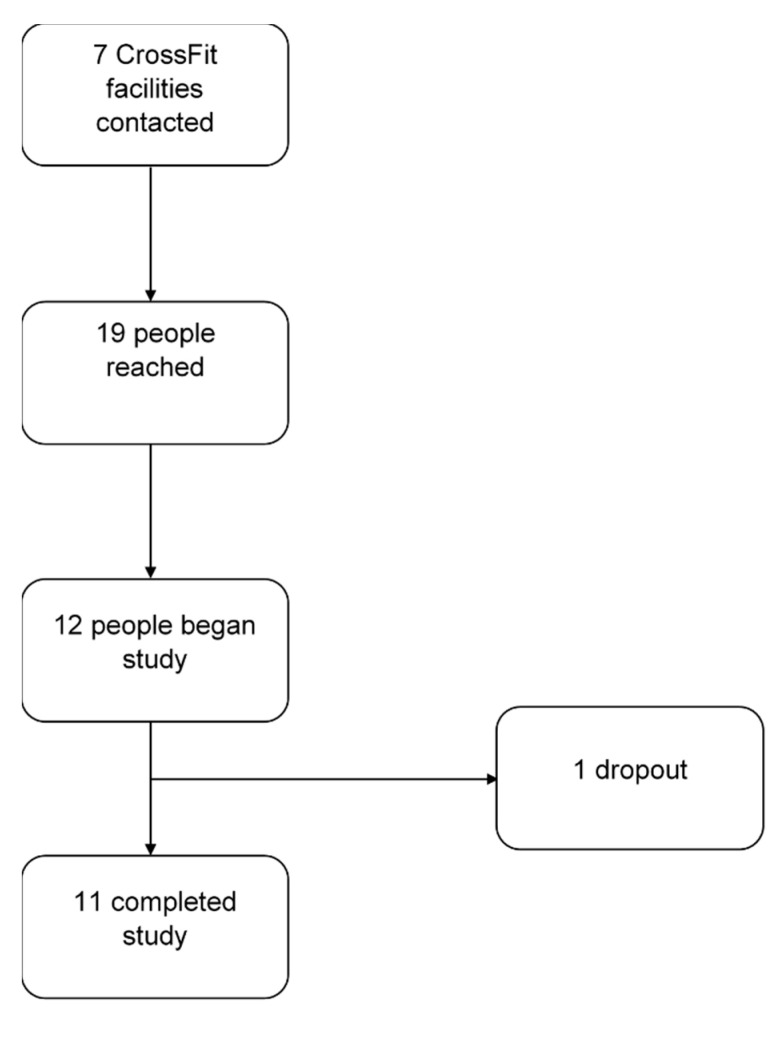
CONSORT diagram illustrating participant recruitment efforts.

**Figure 2 sports-08-00092-f002:**
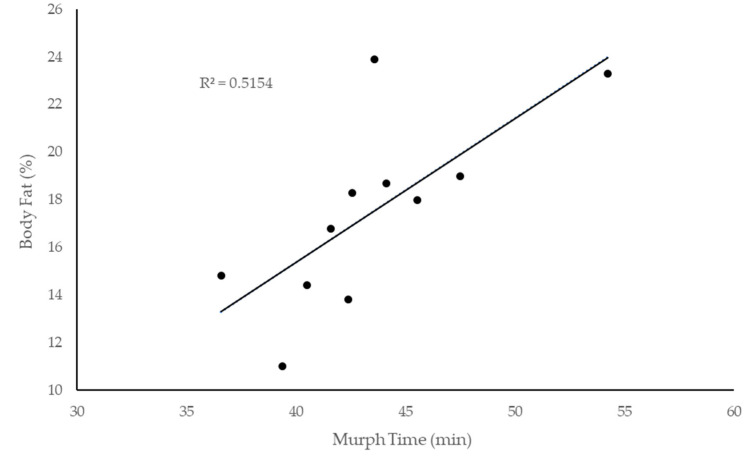
Relationship between body fat percentage and total Murph completion time.

**Figure 3 sports-08-00092-f003:**
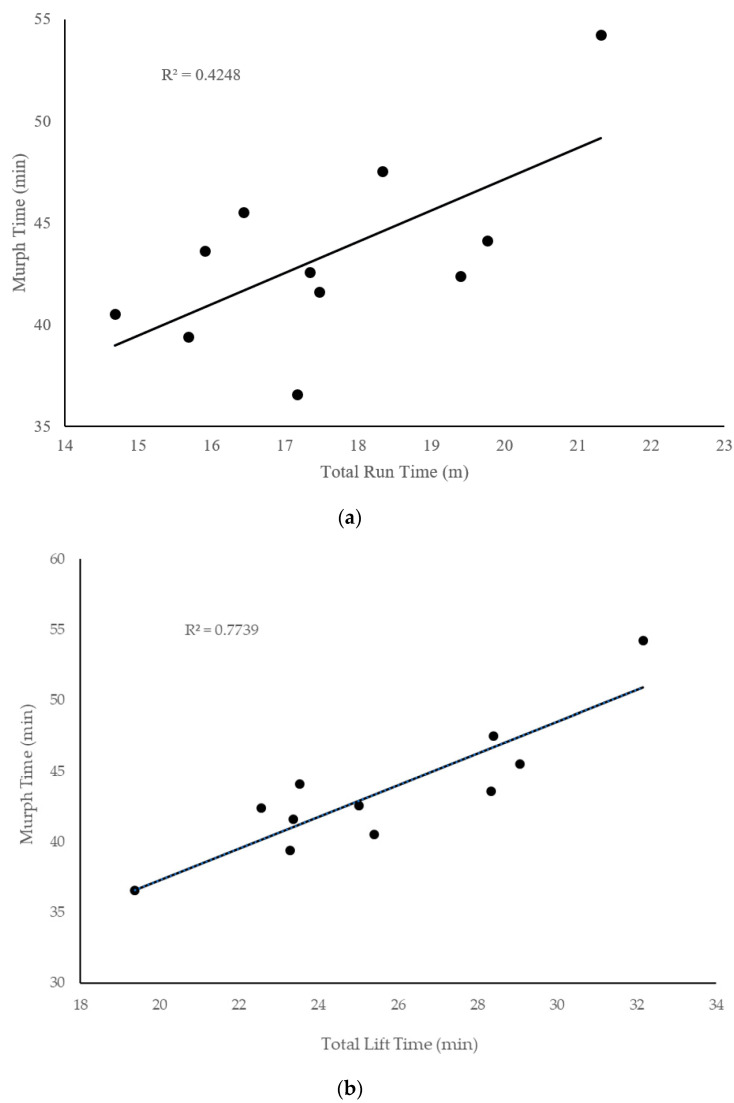
Relationships between total Murph completion time and (**a**) total run time (i.e., run time 1 + run time 2) and (**b**) total lift time.

**Table 1 sports-08-00092-t001:** Participant Characteristics (*n* = 11).

	Mean (±sd)
Age (years)	27.18 ± 3.31
CrossFit Experience (months)	46.82 ± 30.70
Body Mass (kg)	83.32 ± 12.76
BMI (kg/m^2^)	26.74 ± 2.41
Height (cm)	176.09 ± 7.76

**Table 2 sports-08-00092-t002:** Performance Measurements and Correlations with Murph Time.

Physiological Parameter	Mean (±sd)	r-Value	*p*-Value
Body Composition			
Body Fat (%)	17.45 ± 3.90	0.718 *	0.013
Muscular Strength and Endurance			
Strength Total (kg)	267.09 ± 56.57	−0.023	0.947
Relative Strength Total	3.04 ± 0.71	−0.002	0.996
Upper Body Endurance (repetitions)	33.27 ± 5.46	−0.021	0.95
Anaerobic Power			
Peak Power (W)	1054.25 ± 190.63	0.024	0.943
Relative Peak Power (W/kg)	12.42 ± 1.13	−0.347	0.296
Average Power (W)	765.91 ± 128.73	−0.014	0.968
Relative Average Power (W/kg)	9.00 ± 0.70	−0.436	0.180
Anaerobic Fatigue (%)	53.78 ± 6.65	0.165	0.628
Work (kJ)	21.97 ± 3.65	−0.074	0.828
Aerobic Capacity			
VO_2_max (L/min)	4.14 ± 0.87	−0.061	−0.858
VO_2_max (mL/kg/min)	49.52 ± 7.13	−0.423	0.195
Physiological Responses			
Post Lactate (mmol/L)	10.01 ± 3.04	−0.343	0.366
Change in lactate (mmol/L)	7.60 ± 3.50	−0.408	0.316
Max HR (bpm)	185.63 ± 7.64	−0.294	0.381
Average HR (bpm)	168.81 ± 6.41	−0.056	0.871
Murph Subcomponents			
Run 1 (min)	7.46 ± 1.84	0.206	0.544
Run 2 (min)	10.10 ± 1.71	0.509	0.110
Total Run (min)	17.60 ± 1.97	0.652 *	0.030
Lift Time (min)	25.49 ± 3.65	0.880 **	<0.001

Body composition was assessed using dual-energy X-ray absorptiometry (DEXA). Strength Total was calculated as the sum of bench press one-repetition maximum (1RM) and back squat 1RM. Upper Body Endurance was evaluated as maximum amount of repetitions at 50% of bench press 1RM. Anaerobic power was determined by a 30-s Wingate test. Anaerobic Fatigue was calculated as the decline with a higher number indicating greater fatigue. VO_2_ max was performed on a treadmill. * Correlation is significant at the 0.05 level. ** Correlation is significant at the 0.01 level.

**Table 3 sports-08-00092-t003:** Performance Measurement and Correlations with Murph Subcomponents.

Physiological Parameter	Run 1	Run 2	Total Run	Total Lift
Body Composition				
Body Fat (r)	0.337	0.797 **	0.386	0.728 *
*p*	0.31	0.003	0.241	0.011
Muscular Strength and Endurance				
Strength Total (r)	0.242	−0.167	0.051	−0.097
*p*	0.473	0.623	0.881	0.776
Relative Strength Total (r)	0.384	0.155	0.483	−0.32
*p*	0.244	0.648	0.132	0.338
Bench Endurance (r)	−0.526	0.354	−0.188	0.135
*p*	0.096	0.285	0.581	0.693
Anaerobic Power				
Peak Power (r)	0.04	−0.175	−0.138	0.091
*p*	0.907	0.606	0.685	0.791
Relative Peak Power (r)	0.026	−0.23	−0.181	−0.321
*p*	0.941	0.496	0.594	0.336
Average Power (r)	−0.252	−0.143	−0.383	0.208
*p*	0.456	0.674	0.245	0.538
Relative Average Power (r)	−0.48	−0.211	−0.634 *	−0.126
*p*	0.135	0.533	0.036	0.713
Anaerobic Fatigue (r)	0.712 *	−0.022	0.649 *	−0.241
*p*	0.014	0.949	0.031	0.475
Work (r)	−0.252	−0.215	−0.447	0.169
*p*	0.455	0.525	0.168	0.619
Aerobic Capacity				
VO_2_max (r)	−0.246	−0.129	−0.371	0.139
*p*	0.465	0.706	0.261	0.685
Relative VO_2_max (r)	−0.386	−0.214	−0.568	−0.179
*p*	0.241	0.528	0.068	0.599

* Correlation is significant at the 0.05 level. ** Correlation is significant at the 0.01 level.

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
