# Peer review of "Physiological Predictors of Performance on the CrossFit “Murph” Challenge"

_sports, 2020, doi:10.3390/sports8070092_

Round 1
Reviewer 1 Report
Overall, very interesting and novel study. I really enjoyed reading it. I do think you should re-work your introduction, but otherwise the rest of the article is excellent.
- Please work on your introduction. Definition of CrossFit. Whether you use the dictionary, please cite your source. For example, if you look at: https://dictionary.cambridge.org/us/dictionary/english/crossfit
It is different that your definition.
https://www.lexico.com/en/definition/crossfit also is closer but different.
Line 28, if you change "intensive" to "high intensity" then that would get closer to the dictionary definition.
2. Line 31. this is very subjective, please be objective. CrossFit programming consists of "constantly varied, high intensity, functional movements" to improve fitness and health
https://www.crossfit.com/faq
3. Line 32. I don't think the wording is appropriate. consider changing significant amount of stress to a significant "stimulus" to produce adaptations
4. Line 34. name, his last name is spelt wrong it is not "Murphey", it's Murphy.
Reviewer 2 Report
Overall this was a great study that you all should be very proud of. I have participated in the Murph workout several times, which makes me appreciate this study even more. Due to the growing popularity of this workout nation-wide, especially on Memorial Day, it is a very applicable study.
I have a few comments and suggestions for the document, as well as a few questions to address:
- Line 34: Change Murphey to Murphy
- I would suggest placing a space between the number and symbol, for example - changing n=11 to n = 11.
- Why did you all just assess upper extremity muscular endurance, but not lower extremity knowing there are 300 squats in the workout?
- Why did you choose to assess UE muscular endurance with the bench press protocol and not the ACSM push-up protocol?
- Why did you choose to use the 10-20-30 partition?
- I would also address that this is a limitation. Your results are specific to this partitioning - the results may not be the same if the participants did not partition (performed 100, 200, 300 of respective movements), or if partitioned a different way.
- Do you have the data for when participants could no longer complete the 10-20-30 and had to switch to lower reps
- How did you decide when they dropped the number of reps completed/round? Was it as soon as they could not complete all 10, 20, or 30?
- Did they perform kipping pullups or strict?
- I understand the use of the treadmill, and you did mention it in the limitations, but what grade was the treadmill set at? It has been recommended that a 1% grade mimics running outside more than 0% grade.
- I think a limitation could be the amount of assessments performed in the screening battery. The participants' VO2max performance could have been decreased after completing all of the previous assessments, especially the Wingate.
- Had any of the participants completed the Murph workout prior to this study? If so, how many times?
- If so, this could have been a factor in their performance as well.
